# Retrospective record review of pregnant women treated for rifampicin-resistant tuberculosis in South Africa

Martie van der Walt[1]☯, Sikhethiwe Masuku[1]☯*, Sonja Botha[1]☯, Tshifhiwa Nkwenika[2]☯, Karen H. Keddy[1]☯

1 TB Platform Research Unit, Pretoria, South African Medical Research Council, Pretoria, South Africa,
2 Biostatistics Research Unit, Pretoria, South African Medical Research Council, Pretoria, South Africa

☯ These authors contributed equally to this work.
* Sikhethiwe.Masuku@mrc.ac.za, khethi12@gmail.com

**Data Availability Statement:** The data file will be attached as supporting information.

**Funding:** The study was funded by our organization, the South African Medical Research

## Abstract

### Background

Tuberculosis (TB) is amongst the top five causes of death in women of childbearing age (15-≤44 years). Little is known about treatment of pregnant women with drug-resistant TB (DR-TB). Treatment for pregnant women remains challenging and more complex in DR-TB/HIV co-infection, where an evidence-based guide to clinical practice is limited. The study reviewed treatment and pregnancy outcomes and birth outcomes of their new-born in a cohort of pregnant women with DR-TB from three MDR-TB hospitals during 2010 and 2018.

### Design/Methods

Data were extracted from: TB register and patient clinic notes using a standardized case record form. Information on DR-TB treatment, pregnancy and Adverse Drug Events (ADEs) of twenty-six pregnant women treated with individualized second-line TB medications were captured. The frequency of favourable and adverse outcomes regarding disease and pregnancy were evaluated.

### Results

The mean age was 29 years (SD ±5.1), with the minimum and maximum age of 21 and 40 years, respectively. Eleven (42.3%) were previously treated with first-line TB drugs, 11 (42.3%) never treated before and 4 (15.4%) were previously treated for DR-TB. Of the 26 women, 15 (57.7%) had at least one ADE, but most had more than one ADE. Seventeen women were successfully treated, and 22 live births recorded. Live birth outcome was significantly associated with trimester of initiation of DR-TB treatment (p = 0.036). The proportion of live births for the pregnancy trimester when DR-TB treatment was initiated, were 60.0%, 90.9% and 100.0%, for first, second and third trimester, respectively.

Council (SAMRC), TB Platform. The authors are employees of SAMRC therefore, received salaries, travel and accommodation during data collection. "The funders had no role in study design, data collection and analysis, decision to publish, or preparation of the manuscript.

**Competing interests:** The authors have declared that no competing interests exist.

## Conclusion

DR-TB treatment should be delayed until after the first trimester. Routine pharmacovigilance surveillance integrated antenatal and delivery services with an integrated record of DR-TB treatment during pregnancy is recommended. Prospective studies using standardised case record forms for DR-TB treatment for pregnant women could provide more insight on the effect of DR-TB treatment on the birth outcome.

## Background

South Africa has one of the world's most serious tuberculosis (TB) epidemics, that in recent decades has been driven by the human immunodeficiency virus (HIV) epidemic [1, 2]. TB is amongst the top five causes of death in women of childbearing age in the 15-≤44 year group [3]. In 2018, there were an estimated 11,000 Rifampicin (RIF)-resistant TB cases in South Africa, of these an estimated 62% were multidrug-resistant TB (MDR-TB) [1]. MDR-TB is defined as both RIF and isoniazid (INH) resistant, extensively drug-resistant TB (XDR-TB) is defined as MDR-TB and resistance to one second-line fluoroquinolone and one second-line injectable [4].

In 2016 it was estimated that 8,400 pregnant women had drug-susceptible TB in South Africa [5, 6] but little is known about the number of pregnant women with drug-resistant TB (DR-TB) [7]. The treatment remains challenging and is more complex in DR-TB/HIV co-infection where there is a limited evidence base to guide clinical practice. South Africa's estimated population-level HIV prevalence is 13.5% and amongst women in antenatal care 30.8% (30.0%-31.6%) [8, 9]. Since 2013, concurrent ART and TB treatment without considering CD4 count for all HIV infected pregnant women was implemented [10]. Possible adverse effects on the foetus are unknown during DR-TB treatment as pregnant women are commonly excluded from clinical trials [11, 12]. DR-TB treatment in pregnant women present management dilemmas, with care providers suggesting initiation of treatment after the first trimester to protect the foetus from the teratogenic impact [13]. If delay in treatment is not possible, the clinician may recommend termination of pregnancy, especially where the mother's life is at risk due to the severity of the DR-TB or treatment could be teratogenic to the foetus [14, 15]. The top five leading causes of disability and neonatal disorders during 2017 were TB and HIV treatment globally [16].

Reports on pregnancy and treatment outcomes of pregnant women with DR-TB are scarce. Pregnant women are usually excluded from clinical trials and there are limited data available to inform use of DR-TB/HIV treatment in pregnant women. This study reviewed retrospective data from records of pregnant women in South Africa's DR-TB facilities to provide insight into DR-TB treatment outcomes, occurrence of ADEs (including medication errors, adverse drug reactions and allergic reactions) and birth outcomes.

## Methods

### Study design and population

Records of women who were pregnant before or while receiving individualized DR-TB therapy, with pulmonary tuberculosis disease between January 2010 and December 2018 were reviewed. Women were identified by reviewing medical records and interviewing health care providers at the MDR-TB hospitals of Northern Cape, North West, and Free State Provinces.

The Provinces were requested to identify the women from their DR-TB registers for the study period. Women aged between 15 and 44 years with any bacteriological confirmed RIF mono resistance, MDR-TB or XDR-TB and initiated on DR-TB treatment were included in the study. A total 26 records identified by the facility were reviewed.

### Data collection and analysis

Using a standardized case record form, we extracted data from two data sources: the DR-TB register and the patient clinical notes. We collected age and clinical information, including TB history and outcomes, drug-resistance, regimens, ADEs, treatment outcomes and birth outcome. Data were collected from November 2018 to June 2019.

Variables collected were age, birth outcome (defined as live birth, abortion/miscarriage or stillbirth), HIV (infected or uninfected), ART uptake (ART or ART naïve) and DR-TB treatment initiation trimester (first, second or third), DR-TB treatment outcome (cure and treatment completed define as treatment success or lost to follow-up, died and treatment failure outcome) [17]. From clinical notes, adverse drug events were collected.

Data were entered into the password-protected, web-based, certified and accredited RED-Cap version 7.4.1 database and analysed using Stata 15 software. Data analysis included descriptive summary statistics, frequencies, proportions and scores with associated 95% confidence intervals (CI) by the various characteristics of the pregnancy (trimester at initiation of DR-TB treatment, DR-TB type and birth outcome). Fisher's exact test was undertaken to evaluate non-random associations between independent variables and the outcome variables. To establish factors associated with birth outcome (live birth or abortion/miscarriage, stillbirth) and DR-TB treatment outcome (treatment success, died, lost to follow-up or treatment failure) and pregnancy trimester (first, second or third) at the initiation of DR-TB treatment were considered. Confounders were controlled for and a p-value of 0.05 was considered to be statistically significant.

### Ethics approval

Study approved by the Human Research Ethics Committee at the South Africa Medical Research Council (ECO15-9/2019). Waiver of consent was received from provinces.

## Results

Demographic and clinical characteristics of pregnant women are presented in Table 1. The mean age of the women was 29 years (SD ±5.1), with the minimum and maximum age of 21 and 40 years, respectively. The majority were previously treated with first-line TB treatment 11 (42.3%), while 4 (15.4%) were previously treated with DR-TB treatment and 11 had no previous TB history. Fifty percent of women had RIF mono resistant's, 42.3% were RIF & INH resistance, and 7.7% XDR-TB. The drugs used are listed in Fig 1. Most pregnancies were confirmed before initiation of the DR-TB treatment, 12 (46.2%) during the intensive phase of treatment and 7 (26.9%) after the intensive phase.

The prevalence of HIV among the study population was very high, 76.9%. Eighteen HIV infected pregnant women were on ART with less than 10% ART naïve. DR-TB treatment success and favourable birth outcome of live births proportion in this cohort were, 65.4% and 88.0%, respectively (Table 1). There was no statistically significant association between the mother's HIV status and the outcome of the pregnancy (data not shown). Ultimately, whether the mother was tested positive or negative for HIV did not affect the outcome for the fetus. Of the 26 women, 15 (57.7%) had at least one ADE. Adverse drug events ranged from minor to

**Table 1. Demographic and clinical characteristics of pregnant women (N = 26).**

| Variable | Number of pregnant women n (%) | 95% Confidence interval (CI) |
|---|---|---|
| **Age group, years[a]** | | |
| 21–26 | 8 (30.7) | 0.15–0.52 |
| 27–32 | 11 (42.3) | 0.24–0.63 |
| 33–38 | 6 (23.1) | 0.10–0. 44 |
| 39–44 | 1 (3.8) | 0.05–0.25 |
| **One MDR-TB[b] hospital per province** | | |
| Free State Province | 4 (15.4) | 0.55–0.36 |
| North West Province | 15 (57.7) | 0.37–0.76 |
| Northern Cape Province | 7 (26.9) | 0.13–0.48 |
| **Tuberculosis treatment history** | | |
| New- never treated for tuberculosis before | 11 (42.3) | 0.24–0.63 |
| Receive first-line tuberculosis treatment before | 11 (42.3) | 0.24–0.63 |
| Receive drug-resistant tuberculosis treatment before | 4 (15.4) | 0.05–0.36 |
| **Type of drug-resistant tuberculosis** | | |
| Rifampicin mono resistance | 13 (50.0) | 0.30–0.70 |
| Rifampicin including isoniazid resistance | 11 (42.3) | 0.24–0.63 |
| Extensively drug-resistance tuberculosis | 2 (7.7) | 0.02–0.28 |
| **Type of drug-resistant tuberculosis treatment** | | |
| Multidrug resistant tuberculosis | 24 (92.3) | 0.72–0.98 |
| Extensively drug-resistant tuberculosis | 2 (7.7) | 0.02–0.28 |
| **Time of pregnancy confirmation** | | |
| Before drug-resistant tuberculosis treatment initiation | 12 (46.2) | 0.27–0.66 |
| During the intensive phase of treatment | 7 (26.9) | 0.13–0.48 |
| After the intensive phase of treatment | 7 (26.9) | 0.13–0.48 |
| **HIV status and on antiretroviral treatment** | | |
| HIV-uninfected | 6 (23.1) | 0.10–0.44 |
| HIV-infected and on antiretroviral treatment | 18 (69.2) | 0.48–0.85 |
| HIV-infected and antiretroviral treatment naïve | 2 (7.7) | 0.02–0.28 |
| **Drug-resistant tuberculosis treatment outcome** | | |
| Died | 1 (3.8) | 0.05–0.25 |
| Lost to follow-up | 7 (26.9) | 0.13–0.48 |
| Treatment failure | 1 (3.8) | 0.05–0.25 |
| Treatment success | 17 (65.4) | 0.44–0.82 |
| **Birth outcome [n = 25][c]** | | |
| Abortion/miscarriage | 2 (8.0) | 0.02–0.29 |
| Live birth | 22 (88.0) | 0.67–0.94 |
| Stillbirth | 1 (4.0) | 0.05–0.26 |
| **Number of women with adverse events [n = 26]** | | |
| Women without reported adverse events | 11 (42.3) | 0.24–0.63 |
| Women with reported adverse events | 15 (57.7) | 0.40–0.76 |

[a] Mean age 29.2 years.

[b] Multidrug resistant tuberculosis (MDR-TB).

[c] Number reduced- an unknown birth outcome.

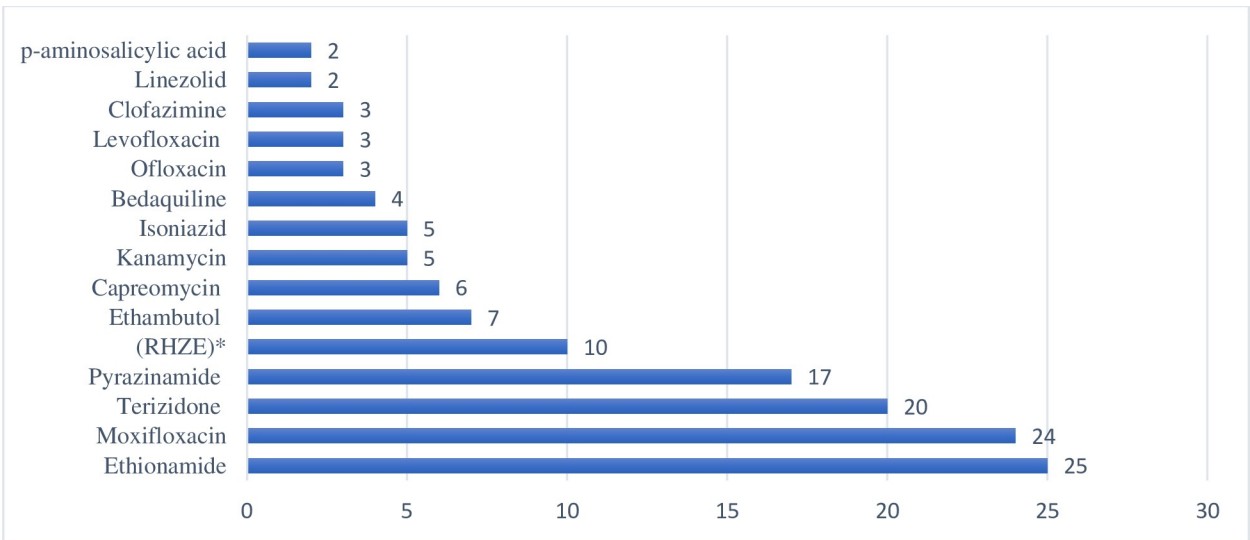

*Fixed dose combination, first-line tuberculosis tablet consists of (rifampicin, isoniazid, pyrazinamide and ethambutol)

**Fig 1. Frequency of tuberculosis drugs used in the study cohort.** *Fixed dose combination, first-line tuberculosis tablet consists of (rifampicin, isoniazid, pyrazinamide and ethambutol).

severe, included systemic and psychiatric disorders and two women had hearing loss however, there was no grading of the adverse events (Fig 2).

After controlling for potential confounders (Table 2), there was no statistically significant association between type of drug-resistant TB, DR-TB treatment HIV and ART, ADE, and birth outcome. However, birth outcome was significantly associated with the trimester in

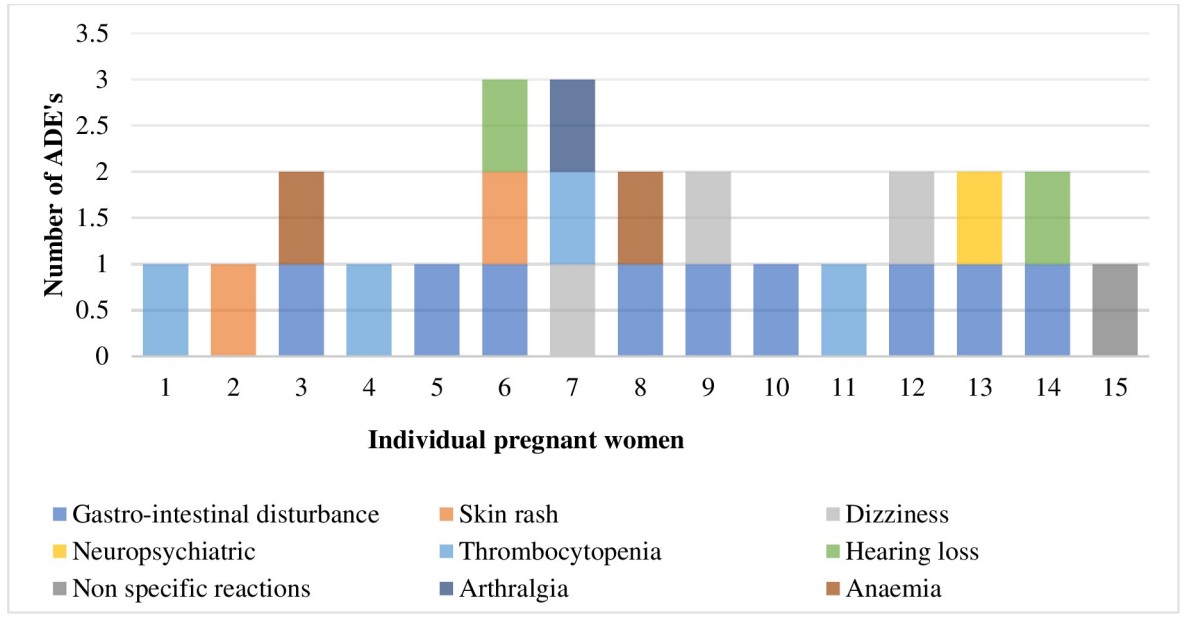

**Fig 2. Pregnant women and the reported adverse drug events (n = 15).**

**Table 2. Proportion of pregnant women with favourable or unfavourable birth outcome (n = 25).**

| Variable | Favourable | Unfavourable | | |
|---|---|---|---|---|
| | Live birth | Abortion/Miscarriage | Stillbirth | p-value |
| | n (%) | n (%) | n (%) | |
| **Type of drug-resistant tuberculosis** | | | | |
| Multidrug resistance tuberculosis | 10 (90.9) | 1 (9.1) | 1 (0.0) | 0.230 |
| Rifampicin mono-resistance tuberculosis | 11 (91.7) | 0 (0.0) | 1 (8.3) | |
| Extensively drug-resistant tuberculosis | 1 (50.0) | 1 (50.0) | 0 (0.0) | |
| **Type of drug-resistant tuberculosis treatment** | | | | |
| Multidrug-resistant tuberculosis | 21 (91.3) | 1 (4.4) | 1 (4.4) | 0.230 |
| Extensively drug-resistant tuberculosis | 1 (50.0) | 1 (50.0) | 0 (0.0) | |
| **Trimester at DR-TB[a] treatment initiate [n = 24][b]** | | | | |
| First trimester | 3 (60.0) | 2 (40.0) | 0 (0.0) | **0.036** |
| Second trimester | 10 (90.9) | 0 (0.0) | 1 (9.1) | |
| Third trimester | 8 (100.0) | 0 (0.0) | 0 (0.0) | |
| **HIV status and antiretroviral treatment** | | | | |
| HIV-uninfected | 6 (100.0) | 0 (0.0) | 0 (0.0) | 1.000 |
| HIV-infected & on ART[c] at DR-TB[a] treatment initiation | 14 (82.4) | 2 (11.8) | 1 (5.9) | |
| HIV-infected & ART[c] initiate, after DR-TB[a] treatment initiation | 2 (100.0) | 0 (0.0) | 0 (0.0) | |
| **Pregnant women with recorded adverse drug events** | | | | |
| Women without recorded adverse drug events | 10 (100.0) | 0 (0.0) | 0 (0.0) | 0.696 |
| Women with recorded adverse drug events | 12 (80.0) | 2 (13.3) | 1 (4.0) | |

[a] Drug-resistant tuberculosis (DR-TB)

[b] Number reduced; one woman's pregnancy trimester unknown

[c] Antiretroviral treatment (ART). Significant values appear in bold.

which a patient was initiated with DR-TB treatment (p = 0.036). The proportion of live births with respect to the stage of pregnancy a patient was initiated with DR-TB treatment, were 60.0%, 90.9% and 100.0%, for first, second and third trimester, respectively.

## Discussion

This study retrospectively reviewed 26 DR-TB treated pregnant women, from a representative MDR-TB hospital from three different provinces of South Africa, who became or were pregnant during DR-TB treatment between the period 2010 and 2018. Of the 26 women, 65.4% had successful DR-TB treatment outcome. Treatment success in this cohort was higher than reported in a similar study from Peru while WHO reported an overall successful treatment outcomes among DR-TB of 54% in 2016 for South Africa [1, 18].

Of these 25 women with the known pregnancy outcome, 88.0% had favourable birth outcomes and 12.0% had unfavourable birth outcomes which included two abortions/miscarriages and one stillbirth. The study could not establish if the abortions were selective or spontaneous as records did not specify. Furthermore, information such as type of delivery, complications of pregnancy, birth weight of the babies and the presence of birth abnormalities were missing in the clinical records.

The trimester when DR-TB treatment was initiated, was significantly associated with birth outcome. Women who initiated DR-TB treatment during the first trimester were more likely to have lower live births, 60.0% than women who initiated DR-TB treatment after the first trimester 90.9%. Initiation during third trimester had 100.0% live births. Similarly, a favourable

birth outcome was also associated with DR-TB treatment initiation during the second trimester in a Ukraine study [19].

Adverse drug events were reported by 57.7% women in this study. However, grading of the adverse events was not recorded in clinical notes. Lack of standardised reporting and recording in pregnant women during DR-TB treatment, contribute to under-reporting on adverse drug events which may have negative impact on foetus and pregnancy outcomes [20].

The findings are subject to the limitations of our study: a retrospective review, the small sample size and that the names of women eligible for inclusion relied on recall by the health providers (i.e., nurses and doctors) and some women could have been omitted. The retrospective method of this study could not establish if abortions were elective or spontaneous as this was not recorded in the clinical notes. Similarly, abnormal live births or birth defects amongst live births could not be established as DR-TB and maternal clinical records are not integrated. The small sample size and lack of control comparison limit our ability to test for an association between the use of second-line TB drugs and outcomes of treatment or pregnancy. A similar limitation was documented in a Peru study [18]. Lack of standardised recording, case report forms and reporting leads to poor identification of the implicated drug for adverse events. Not all women were tested for pregnancy prior to initiation of DR-TB treatment in this study. Our findings concur with a similar study done in Eastern Europe study on MDR-TB treatment during pregnancy [21].

## Conclusion

An informed right to choose between voluntary termination of pregnancy or continuation with potential teratogenic treatment should be given to pregnant women if diagnosed with RIF resistant TB before DR-TB treatment is initiated. DR-TB treatment initiation should be delayed until the second but preferably the third trimester. This study found that delaying of DR-TB treatment initiation until the second trimester was associated with more favourable birth outcomes. Routine pharmacovigilant surveillance of DR-TB treatment during pregnancy is recommended. Routine, nationally standardised data recording of favourable and unfavourable birth outcomes attributed to DR-TB and/HIV treatment and indicating if abortions are elective or spontaneous is recommended. Further research on DR-TB treatment for pregnant women using standardised protocols and case report forms is advocated.

## Supporting information

**S1 File. Tb women 1 Sep 2020.**
(DTA)

## Acknowledgments

The authors gratefully acknowledge the contribution of Dr H. Ferreira, North West Department of Health, Dr B. Mastrapa, Northern Cape Department of Health and Dr N M Zakhura, Free State Department of Health.

## Author Contributions

**Conceptualization:** Martie van der Walt, Sikhethiwe Masuku, Sonja Botha, Tshifhiwa Nkwenika.

**Formal analysis:** Tshifhiwa Nkwenika.

**Methodology:** Sikhethiwe Masuku, Sonja Botha.

**Writing – original draft:** Sikhethiwe Masuku, Sonja Botha.

**Writing – review & editing:** Martie van der Walt, Sikhethiwe Masuku, Sonja Botha, Tshifhiwa Nkwenika, Karen H. Keddy.

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
