## [Decision Letter · Decision Letter 0]

20 Jul 2020

PONE-D-20-19362

Retrospective record review of pregnant women treated for rifampicin-resistant tuberculosis in South Africa

PLOS ONE

Dear Dr. Masuku,

Thank you for submitting your manuscript to PLOS ONE. After careful consideration, we feel that it has merit but does not fully meet PLOS ONE’s publication criteria as it currently stands. Therefore, we invite you to submit a revised version of the manuscript that addresses the points raised during the review process.

We look forward to receiving your revised manuscript.

Kind regards,

Hasnain Seyed Ehtesham

Academic Editor

PLOS ONE

Additional Editor Comments:

Minor Revision

Journal Requirements:

2. In your Methods section please include the dates upon which authors accessed the data sources employed in this study.

5. Please upload a copy of Figure 2, to which you refer in your text on page 7. If the figure is no longer to be included as part of the submission please remove all reference to it within the text.

Reviewers' comments:

Reviewer's Responses to Questions

**Comments to the Author**

1. Is the manuscript technically sound, and do the data support the conclusions?

Reviewer #1: Yes

Reviewer #2: Yes

2. Has the statistical analysis been performed appropriately and rigorously? 

Reviewer #1: Yes

Reviewer #2: Yes

3. Have the authors made all data underlying the findings in their manuscript fully available?

Reviewer #1: Yes

Reviewer #2: Yes

4. Is the manuscript presented in an intelligible fashion and written in standard English?

Reviewer #1: Yes

Reviewer #2: Yes

5. Review Comments to the Author

Reviewer #1: Walt et al, nicely reported on pregnancy and treatment outcomes of pregnant women with DR-TB among South-Africa population.

Following suggestive comments would make the manuscript stronger and beneficial for the readers:

Abstract:

• Word limits excided as per Journal guidelines. Should be modified.

• While considering the word limit, author should write about the total number of patients screened and enrolled in study.

Method:

• If possible, author should specify the type of TB infection (PTB or EPTB) among patients.

• Authors should present sociodemographic data as maintained the method section.

Results:

Table 1: Age group, Years; 15-20. Author should remove the age limit because of no patients were enrolled in this group.

Reviewer #2: This retrospective study provides information regarding Anti tuberculous treatment outcome on pregnant women. However as the data suggests HIV positive percentage among the cohort was high, its correlation with the study outcome on successful birth is not briefed in the present manuscript. This point needs to be highlighted in the revised manuscript.

6. PLOS authors have the option to publish the peer review history of their article (what does this mean?). If published, this will include your full peer review and any attached files.

Reviewer #1: No

Reviewer #2: No

---

## [Author Response · Author response to Decision Letter 0]

21 Aug 2020

Response to Reviewers Comments

Reviewer #1: Walt et al, nicely reported on pregnancy and treatment outcomes of pregnant women with DR-TB among South-Africa population:

Following suggestive comments would make the manuscript stronger and beneficial for the readers:

Abstract: 

• Word limits excided as per Journal guidelines. Should be modified.

Response: Abstract revised to 300 words

• While considering the word limit, author should write about the total number of patients screened and enrolled in study.

Response: The facilities were requested to identify the records for women who were or got pregnant during DR-TB treatment. A total 26 records identified by the facility were reviewed. (See line 76)

Method:

• If possible, author should specify the type of TB infection (PTB or EPTB) among patients.

Response: Records of women who were pregnant before or while receiving individualized DR-TB therapy, with pulmonary tuberculosis disease between January 2010 and December 2018 were reviewed. (See line 70-71)

• Authors should present sociodemographic data as maintained the method section.

Response: Sentence corrected. We collected sociodemographic age and clinical information, including TB history and outcomes, drug-resistance, regimens, ADEs, treatment outcomes and birth outcome. (See line 79-80)

Table 1: Age group, Years; 15-20. Author should remove the age limit because of no patients were enrolled in this group.

Response: Age group removed. (See Table 1, line 108)

Reviewer #2: This retrospective study provides information regarding Anti tuberculous treatment outcome on pregnant women. However, as the data suggests HIV positive percentage among the cohort was high, its correlation with the study outcome on successful birth is not briefed in the present manuscript. This point needs to be highlighted in the revised manuscript.

Response: The pregnancy outcome was not statistically significantly associated with HIV-status of the mother (data not shown). Ultimately, whether the mother was tested positive or negative for HIV did not affect the outcome for the fetus. (See line 117-120)

6. PLOS authors have the option to publish the peer review history of their article (what does this mean?). If published, this will include your full peer review and any attached files.

---

## [Editor Report · Decision Letter 1]

28 Aug 2020

Retrospective record review of pregnant women treated for rifampicin-resistant tuberculosis in South Africa

PONE-D-20-19362R1

Dear Dr.  Masuku,

We’re pleased to inform you that your manuscript has been judged scientifically suitable for publication and will be formally accepted for publication once it meets all outstanding technical requirements.

Kind regards,

Hasnain Seyed Ehtesham

Academic Editor

PLOS ONE

Additional Editor Comments (optional):

I have gone through this manuscript and also the Authors response to the comments of both the reviewers. This manuscript was sent for minor revision and Authors have made the necessary changes addressing the comments of reviewers. Abstract of the manuscript has been revised as per journals guidelines and age group has been also removed. The Author's explanation to the comments of the 2nd reviewer is quite satisfactory. I recommend this manuscript for publication.
---

## [Editor Report · Acceptance letter]

15 Sep 2020

PONE-D-20-19362R1 

Retrospective record review of pregnant women treated for rifampicin-resistant tuberculosis in South Africa 

Dear Dr. Masuku:

I'm pleased to inform you that your manuscript has been deemed suitable for publication in PLOS ONE. Congratulations! Your manuscript is now with our production department. 

Kind regards, 

on behalf of

Prof Hasnain Seyed Ehtesham 

Academic Editor

PLOS ONE